# Comparing Methods for Segmenting Elementary Discourse Units in a French Conversational Corpus

**Laurent Prévot**
Aix Marseille Université & CNRS
Laboratoire Parole et Langage
Aix-en-Provence, France
laurent.prevot@univ-amu.fr

**Julie Hunter**
LINAGORA Labs
Toulouse, France
hunter@linagora.com

**Philippe Muller**
Toulouse Université & CNRS
IRIT
Toulouse, France
philippe.muller@irit.fr

## Abstract

While discourse segmentation and parsing has made considerable progress in recent years, discursive analysis of conversational speech remains a difficult issue. In this paper, we exploit a French data set that has been manually segmented into discourse units to compare two approaches to discourse segmentation: fine-tuning existing systems on manual segmentation vs. using hand-crafted labeling rules to develop a weakly supervised segmenter. Our results show that both approaches yield similar performance in terms of f-score while data programming requires less manual annotation work. In a second experiment we play with the amount of training data used for fine-tuning systems and show that a small amount of hand labeled data is enough to obtain good results (albeit not as good as when all available annotated data are used).

## 1 Introduction

Discourse parsing is the decomposition of texts or conversations in functional units that encode participants intentions and their rhetorical relationships. Segmentation in these units is the first step for other levels of analysis, and can help downstream NLP tasks.

Discourse parsing involves determining how each part of a discourse contributes to the discourse as a whole—whether it answers a question that has been asked, provides an explanation of something else that was said, or signals (dis)agreement with a claim made by another speaker. The first step, then, is to decompose a discourse into minimal parts that can serve such discursive functions. We will use the term *elementary discourse unit* (EDU; Asher and Lascarides, 2003) to designate a minimal speech act or communicative unit, where each EDU corresponds roughly to a clause-level content that denotes a single fact or event. While EDU segmentation of written documents has received a lot of attention from the discourse and NLP community, this is less true for segmentation of conversational speech. Conversational data has mostly been approached from either (i) a dialogue act segmentation and tagging perspective, and usually on rather task-oriented dialogues (Dang et al., 2020) or (ii) punctuation prediction to enrich transcripts obtained with Automatic Speech Recognition (Batista et al., 2012).

We assume that this situation encourages a bias toward written genres that can be problematic for discourse segmentation, because those genres (newspapers, literature,...) tend to include long complex sentences, while actual conversations are made of relatively short contributions. Non-sentential units (Fernández et al., 2007), which often consist of only a single word, are extremely frequent in conversation and can convey full communicative acts, such as an answer to a question or a communicative feedback, that are crucial for modeling discourse structure.

In this paper we benefit from a fully segmented corpus, the Corpus of Interactional Data (CID; Blache et al., 2017), for running a set of experiments on discourse segmentation. This data set is challenging as it consists of 8 long conversations (1 hour each) alternating between interactive narrative sequences like (1) (with a clear dominant speaker holding long turns made of many discourse units) and more dialogical sequences like (2) (made of very short, very often incomplete, turns

that sometimes need to be grouped together to form a valid discourse unit).

(1) [on y va avec des copains]$_{du}$ [on avait pris le ferry en Normandie]$_{du}$ [puisque j'avais un frère qui était en Normandie]$_{du}$ [on traverse]$_{du}$[1]
[we went there with friends]$_{du}$ [we took the ferry in Normandy.]$_{du}$ [since I had a brother who was in Normandy]$_{du}$ [we cross]$_{du}$

(2) A : tu vois à peut près où ///c'est///[2] you know more or less where ///it is///
B: ///oui///
///yes///
A: ouais
yeah

EDUs have become a central topic for discourse parsing (Zeldes et al., 2019a), and in this paper we present two main experiments designed to (i) compare EDU segmentation approaches on our challenging data set and (ii) evaluate the impact of the amount of hand-labeled data used for training. More precisely we first compare the results of several baselines with (i) *state-of-the-art* level segmentation systems fine-tuned on the CID and (ii) a weakly supervised approach bootstrapped by hand crafted labeling functions. Our second experiment consists in varying the amount of hand-labeled data used either for training the base model from scratch or for fine-tuning an existing written text segmenter.

## 2 Previous and Related work

### 2.1 Discourse Segmentation

Discourse segmentation had been largely neglected by work in discourse parsing, and mostly applied to English corpora (Wang et al., 2018), until a few years ago when the multilingual, multi-framework `disrpt` campaigns were introduced (Zeldes et al., 2019b, 2021a). The present paper relies heavily on the French model, Tony (Muller et al., 2019), from those campaigns. Built on the Allen NLP library (Gardner et al., 2018), Tony is a sequential tagging model over contextual embeddings,

namely multilingual BERT (Devlin et al., 2018), that treats segmentation as a token classification problem where each token is either a beginning of a segment or not.

While the overall best segmenter on the `disrpt` tasks is currently Gessler et al. (2021), this segmenter requires syntactic parsing, which is unreliable for highly spontaneous conversational speech of the kind in CID. Moreover, for the sake of our experiment, simple pipelines (based on plain text for Tony) are preferable to more sophisticated ones. Finally, Tony is on par with the best performing French model? (Bakshi and Sharma, 2021). See (Zeldes et al., 2021b) for details on the performance of existing systems.

Gravellier et al. (2021) adapted Tony to conversational data by (i) fine-tuning it on a conversational data set and (ii) adopting a data-programming approach similar to what we propose here. However, the transcriptions used in their work were obtained from a recording setting with a unique microphone. In CID, each participant is recorded on a separate channel, and the transcription of the corpus was fully manual and even corrected several times to reach very high transcription accuracy for a conversational corpus. Moreover, only a small portion of the corpus used by Gravellier et al. (2021) contains gold EDU segmentation ($\approx$ 1100 units), as this corpus was segmented to train a weakly supervised labeling model guided by hand-crafted labeling rules. The present work is grounded on a completely different data set, CID (Blache et al., 2009), that has been fully manually segmented. This corpus provides over 17000 discourse units to experiment with, which allows us to evaluate the quantity of supervised data that is needed to equal or improve performance over the weakly supervised model.

### 2.2 Weak supervision

For the weak supervision part of our experiments, we rely on the so-called "data programming" approach proposed by Ratner et al. (2017). The general principle is to design multiple overlapping heuristic rules for a classification problem, then aggregate them statistically to automatically produce noisy labels on unannotated data that can then be

---

[1]Color alternation is used to highlight discourse units.

[2]///***/// indicates overlapping speech.

fed to a regular supervised model.

This approach has been implemented in the Snorkel library (Ratner et al., 2017) and also independently adapted in the Skweak framework of Lison et al. (2021) and the Spear library (Abhishek et al., 2022). These frameworks provide both an API to define heuristic rules and an aggregation model for the rules. Their output is a noisily annotated data set that can then be used to train the supervised model of one's choice. This approach has been used in discourse analysis to enrich a discourse parser (Badene et al., 2019), and is also the basis of the work in Gravellier et al. (2021) mentioned above. For our final supervised model, we adopted the architecture of Gravellier et al. (2021), but trained it on a different noisy data set.

As explained above, the general idea behind data programming is to leverage expert knowledge by writing a set of labeling functions (*LF*) that can be developed and tested over a very small amount of annotated development data. The system builds a profile for each (*LF*) and a model is trained by combining all *LFs* (LFs being weighted by their accuracies). This model is then used for labeling a training set and finally a supervised model is trained on the data set that had been automatically annotated by the label model.

These frameworks leave open the choice of the final supervised model, since their output is just a (noisily) annotated data set. As the final supervised model, we used the same architecture as previously mentioned work on segmentation Gravellier et al. (2021), but only train it on the noisy data set.

## 3 Gold EDU segmentations

The Corpus of Interactional Data (CID) (8 dyadic conversations, 1 hour duration for each) (Blache et al., 2009, 2017) was segmented following guidelines designed for written documents (Muller et al., 2012) that were modified for spoken conversational data. These guidelines thus combine semantic and discourse criteria (used in particular in monological sequences like (1)) with dialogical and interac-

tional ones (that are more useful in dialogical sequences like (2)). The CID displays highly spontaneous data with colloquial sequences like (3) or strong disfluencies like (4) making discourse segmentation a much more difficult task than on written genres, even for humans. The whole data set consists of about 125 000 tokens for 15,463 discourse units (12,4% of the tokens are EDU boundaries). EDUs are obtained from at least two manual annotations (obtained from 4 naive coders and 2 experts). The mean Cohen's $\kappa$-score across speaker for naive coders is 0.85 (min: 0.83 ; max :0.87). Annotations were performed with Praat (Boersma, 2002) in order to have access to signal word-alignment when making segmentation decisions. The discourse annotations (Prévot et al., 2021) are available from Ortolang platform : `https://www.ortolang.fr/market/item/ortolang-000918`.

(3)   A: [comme ça # ah ouais non c'était]$_{du}$
       A: [like that # oh yeah no it was]$_{du}$
       B: [ah ouais profitez profitez de vos soirées]$_{du}$
       B: [oh yeah enjoy enjoy your evenings]$_{du}$
       A: [ouais c'est pour ça]$_{du}$
       A: [yeah it's for that]$_{du}$

(4)   [ou des euh non pas des f- pas des frustrations]$_{du}$ [des # espèces de euh # mhm # ouais des des vues différentes sur le boulot quoi]$_{du}$
       [or some uh no not some f- not some frustrations]$_{du}$ [some kind of uh # mh # yeah some some different views about work what]$_{du}$

## 4 Method

In this work we use the existing implementation of Tony (Muller et al., 2019) and that of Gravellier et al. (2021), called `tony-w(ritten)` and `tony-s(poken)`, respectively, as baselines. Our first experiment consists in comparing a supervised model (fine-tuning 'Tony' baselines with our annotated data) against the weakly supervised data-programming approach. In a second experiment, we explore the impact of the amount of data used for fine-tuning the models.

Transcripts from the CID do not include any

| name | Polarity | Coverage | Overlaps | Conflict | Correct | Incorrect | Accuracy |
|---|---|---|---|---|---|---|---|
| tony_written | 0,1 | 1.000 | 0.976 | 0.0632 | 102446 | 8994 | 0.919 |
| no_pause | 0 | 0.898 | 0.898 | 0.0474 | 94706 | 5415 | 0.946 |
| long_pause | 1 | 0.054 | 0.054 | 0.0094 | 5353 | 683 | 0.887 |
| very_long_pause | 1 | 0.032 | 0.032 | 0.0051 | 3399 | 162 | 0.955 |
| extreme_pause | 1 | 0.022 | 0.022 | 0.0037 | 2357 | 48 | 0.980 |
| pause_begin_pos | 1 | 0.042 | 0.042 | 0.0101 | 4362 | 328 | 0.930 |
| pause_ending_pos | 1 | 0.036 | 0.036 | 0.0097 | 3338 | 652 | 0.837 |
| non_ending_tok | 0 | 0.323 | 0.323 | 0.0009 | 35579 | 365 | 0.990 |
| pause_begin_tok | 1 | 0.055 | 0.055 | 0.0111 | 5533 | 617 | 0.900 |
| pause_ending_tok | 1 | 0.005 | 0.005 | 0.0005 | 550 | 22 | 0.962 |
| dm_bi_ini | 1 | 0.012 | 0.012 | 0.0059 | 1146 | 222 | 0.838 |
| non_begin_tok | 0 | 0.005 | 0.005 | 0.0001 | 530 | 9 | 0.983 |
| feedback_cluster | 0 | 0.026 | 0.026 | 0.0012 | 2763 | 87 | 0.969 |
| repeat | 0 | 0.088 | 0.088 | 0.0124 | 8678 | 1141 | 0.884 |
| filled_pause | 0 | 0.071 | 0.071 | 0.0055 | 7428 | 570 | 0.929 |
| truncated_word | 0 | 0.016 | 0.016 | 0.0013 | 1629 | 169 | 0.906 |

Table 1: Labeling Function statistics

kind of punctuation ((punctuating conversational speech was taken to be a complex pragmatic annotation task that relies on prosody and other sources of information that are not part of the transcription process). Punctuation, however, is a crucial cue for existing discourse segmenters based on written text. We therefore decided to introduce breaks by treating all pauses in our experiments that were over 200 ms as introducing commas in the token sequence, and all pauses over 900 ms as indicators of "document separation" (like a period in written text). This allowed to help the baseline models trained on textual data and written genres. The idea behind such a short (200 ms) pause duration is that pauses signal places in which a discourse segmentation is likely to happen. When facing these pause/comma tokens, the systems then try to distinguish those corresponding to discourse breaks from the other ones. This does not mean that the system does not predict discourse boundaries at other locations.

### 4.1 Fine Tuning

Fine-tuning of both `tony-w` and `tony-s`– where the latter results from fine-tuning the former with data from a conversational corpus using the data programming approach– proceeded in the same fashion. We first continued to train the original models with the same configurations but with CID labeled data. We conducted a cross-validation experiment in which 7 conversations (7 hours) are used for fine-tuning both models and tested on the remaining eighth conversation, and performed a permutation to obtain a cross-validation.

### 4.2 Data-Programming

Like Gravellier et al. (2021), we pulled from our knowledge of conversational French to define hand-crafted rules (i.e. labeling functions) for the data programming approach. Our approach differed in important ways from that of Gravellier et al. (2021), however, stemming in part from difference in the target data sets and also preprocessing choices. While (Gravellier et al., 2021) attempted to exploit more prosodic and acoustic information in their labeling functions, our rules are based solely on time-aligned (at token level) transcription, NLP annotations (POS-tagging) and duration (in particular pause duration). We also opted for a different POS-tagger: while (Gravellier et al., 2021) used Spacy (Honnibal and Montani, 2017) we chose Stanza (Qi et al., 2020) because it offers a 'spoken' model for French which proved to be more reliable than Spacy for tagging crucial tokens specific to conversational speech. Both the original Tony model and the model developed by Gravellier et al are used to define heuristic LFs.

Table 1 presents the most important labeling functions (LF) retained after various experiments on the development set. The columns of this table are the ones produced by the *La-*

```python
@labeling_function()
def long_pause(x):
    return BEG if (x["prev-tok"]=='#' and x['prev_dur'] > LONG_PAUSE) else ABSTAIN

@labeling_function()
def non_ending_tok(x):
    return NOBEG if x["prev-tok"] in NON_ENDING else ABSTAIN

@labeling_function()
def pause_and_ending_pos(x):
    if ((x["prev-tok"]=='#') and (x["prev_dur"] > PAUSE)
            and (x["pprev-pos"] in ENDING_POS)):
        return BEG
    else:
        return ABSTAIN

@labeling_function()
def repeat(x):
    return NOBEG if x["tok"] in [x["prev-tok"],x["pprev-tok"],x['ppprev-tok']]
                                        else ABSTAIN

@labeling_function()
def filled_pause(x):
    return NOBEG if ((x["prev-tok"] in FP) or (x["tok"] in FP))  else ABSTAIN

@labeling_function()
def truncated_word(x):
    if (str(x["prev-tok"])[-1]=='-'):
        return NOBEG
    elif ((str(x["pprev-tok"])[-1]=='-') and (x["prev-tok"] in [',','*','euh'])):
        return NOBEG
    else:
        return ABSTAIN
```

Figure 1: Labeling Function examples

*beling Function Analysis* function provided by Snorkel: *Polarity* states whether the LF labels a boundary or the absence of a boundary; *Coverage* corresponds to the percentage of instances for which the LF was triggered; *Overlaps* quantifies the proportion of times other LFs are firing at same time as a given LF; *Conflict* quantifies whether any other LFs predict a different label; *Correct/Incorrect* is the amount of correct/incorrect labels based on the development data set and this also defines Accuracy.

Unsurprisingly, `tony` contributes significantly to the prediction of segment bound-

aries. Rules based on pause duration (e.g. `long_pause` in Figure 1) and POS also had a considerable impact on results as did lists of tokens associated with the presence or absence of EDU boundaries. NON-ENDING TOKENS, for example, include various pronouns, determiners, prepositions, negations and initiating discourse markers (See full list in the Appendix). Most of the selected rules involving tokens and POS use pauses as additional criteria (`pause_ending_pos`). The rules `repeat`, `filled_pause` and `truncated_word` target disfluencies, which are generally associated with the absence of an EDU bound-

ary. Finally `feedback_cluster` targets sequences of acknowledgement tokens that generally constitute a single EDU *(e.g., ah ouais d'accord/oh yeah right)*.

The main interest of data-programming is to aggregate sources of information to perform the classification task. For our purposes, the core idea was to combine text-based existing segmentation models with conversational/spoken expert knowledge expressed via labeling functions, and thus our discussion in Section 5 focuses on results the include `tony`. We note, however, that it is possible to compare the results with data-programming models that do not rely on an existing text-based segmentation model. These models tend to have much higher precision ($> 0.75$) but low recall ($< 0.6$) and overall, a lower f-score ($\approx 0.67$). This is due to the fact that the expert LFs are rather precise but fail to cover many cases common to monological sequences in conversation and monologue in text, where `tony` excels. On the other hand, `tony` tends to predict too many boundaries, leading to the drop in precision observed when its predictions are taken into account.

### 4.3 Amount of labeled data

We experimented on varying the amount of labeled data used to train the supervised model (10, 20, 30, 50 or 80%). This was done either as fine-tuning of `tony-w (ft)` or as direct training from the base model (`no-model`) which is a simple BERT model in our case.

## 5 Results and Discussion

### 5.1 Fine-tuning vs. Data programming

The results of the baselines, fine-tuning and the data-programming approach constitute a global coherent picture. Tony baselines show a high recall (Figure 2) but with a relatively low precision (Figure 3). `tony-spoken` starts out significantly worse than `tony-written`. This is probably due to (i) the relative low quality of the transcriptions used for training `tony-spoken` and perhaps the nature of our data which is conversational but hosts a significant amount of narrative sequences.

The baselines based on pause duration only (we show here pause baselines at 200ms and

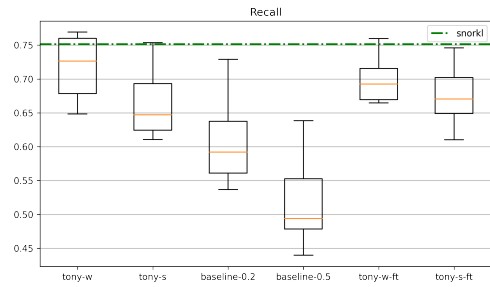

Figure 2: Boundary recall for various configurations; green dashed line = Data-programming

500ms) exhibit a surprisingly high precision, showing the relevance of using this cue as a signal for discourse units. They do miss quite a few cases but overall perform well (especially with a threshold of 200ms). The missing boundaries are discourse units not separated by any pauses, like in (1) for example.

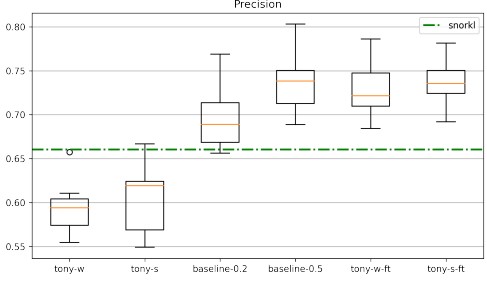

Figure 3: Boundary Precision for various configurations, green dashed line = Data-programming

Fine-tuning really helps `tony` models: recall remains high and precision increases significantly (Figure 4). Fine-tuning allows the model to distinguish which commas (pauses) do not introduce discourse segments.

The comparison of f-scores (Figure 4) of fine-tuning and data-programming approaches does not yield significant differences. It seems to validate the interest of the weakly supervised data programming approach since writing the labeling rules requires much less effort than manually segmenting a large corpus.

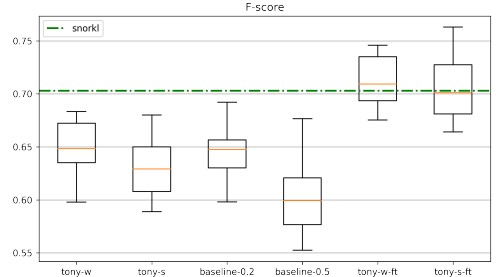

Figure 4: Boundary F-score; green dashed line = Data-programming

## 5.2 Amount of hand labeled data

In our second experiment, we incrementally reduced the amount of annotated data used to train a supervised model in order to determine whether performance would decrease strongly if only a small amount of annotated data were provided.

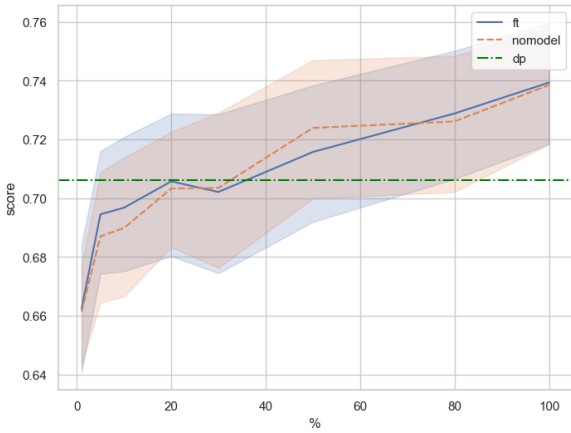

Figure 5: F-score for the supervised models trained on 1, 5, 10, 20, 30, 50, 80% of the original training data set (nomodel: just base BERT, ft: fine-tuned). We also indicate the score of the data programming model (dp). Bands are 95% confidence interval based on the cross-validation.

The results presented in Figure 5 suggest that even if more data is better, a small amount of training data (here 10% corresponds to about 1500 discourse units which is still a significant annotation effort) is enough to really improve the base model. This result mitigates the previous finding: since efficiently fine-tuning existing models does not seem to

require annotating a lot of data, the difference in terms of efficiency between hand-labeling and developing a set of labeling functions is not huge, suggesting that both approaches are worth exploring depending on the use case.

## 5.3 Error Analysis

Error analysis of both data programming and the fine-tuned models yields interesting observations. As expected, phenomena that are really specific to conversational speech are the main sources of errors. For example 'quoi'/'what' is a very common French function word that is heavily used in final position in conversational speech (with a rather unclear function) (Delafontaine, 2020). This item was a major source of error.

Relative clauses anchored on extremely light hosts were also problematic, particularly when they had a restrictive function as in (5). The data-programming approach tended to segment after relative pronouns whether they introduced restrictive or non-restrictive clauses, which generated some over-segmentation.

(5)     [genre des gens || qui étaient au même
        niveau que moi]$_{du}$
        [like people || that were at the same
        level as me]$_{du}$[3]

Another important source of error was complex disfluencies involving discourse markers as illustrated in (6).

(6)     [mais là # || mais euh || mais là c' est
        normal]$_{du}$
        [but in that case ||# but uh || but in
        that case it is normal]$_{du}$

The fine-tuned model introduced errors of its own. It did not segment on certain discourse marker cues like 'mais'/'but' and 'si'/'if'. It does not seem to judge them to be reliable initiators of discourse units.

A second source of error for this model was the repetition of presentative constructions 'c'est'/'it is' of which an extreme example is given in (7).

---

[3]In the error analysis examples, || stands for a false positive (added a boundary in a wrong place) and $$ for false negative (missed a boundary).

(7)   [non c' est plus de la recherche]$_{du}$ $$
      [c' est de la c' est de la # c' est #
      a-]$_{du}$ # [ouais voilà]$_{du}$ # [c' est de la
      t- c- c' est]$_{du}$ $$ # [co- comment ça
      s' appelle]$_{du}$ $$ [c' est de la # || de la
      capitalisation]$_{du}$
      [no it is not research anymore]$_{du}$ $$ [it
      is some it is some # it is # a-]$_{du}$ #
      [yeah that's right]$_{du}$ [# it is some t- c-
      it is h-]$_{du}$ # [how do you call it]$_{du}$ $$ [it
      is some # some capitalisation]$_{du}$

There are a wide range of other errors represented, though they are less frequent. They include (i) long pauses (>1s) that do not actually split a discourse unit. As explained above, our preprocessing step splits 'documents' based on pauses that last more than 900ms, and while during fine-tuning, the models see that a 'document start' does not always correspond to a 'discourse unit start', document starts tend to be used for detecting boundaries (because they are always in the model before fine-tuning); (ii) dialogical structures (involving both speakers) that are currently not handled; (iii) reported speech (that was systemically segmented in the manual annotation even if sometimes the reported speech introduction was extremely light in content).

In the CID, there are two kinds of sequences: (i) narrative sequences in which one of the participants tells a story (with an interactive flavor involving feedback and production from the other participant but in which there is a clear main speaker and a narrative flow), and (ii) transition sequences where the participants comment and chat about these stories, as well as negotiate who will tell the next story and what it will be about. As expected, narrative sequences are better handled by our models, even when produced at a relatively fast pace that did not allow for pauses between discourse units like (8) which is the continuation of our example (1) and in which there are no pauses (longer than 200 ms) but several discourse units.

(8)   [on y va avec des copains]$_{du}$ [on avait
      pris le ferry en Normandie]$_{du}$ [puisque j'
      avais un frère qui était en Normandie]$_{du}$

[on traverse]$_{du}$ [on avait passé une nuit
épouvantable sur le ferry]$_{du}$
–
[we went there with friends]$_{du}$ [we took
the ferry in Normandy.]$_{du}$ [since I had
a brother who was in Normandy]$_{du}$ [we
cross]$_{du}$ [we spent a terrible night on
the ferry]$_{du}$

However, even in narrative sequences some common spoken constructions seem to cause problems for the models, including presentatives such as *y a/y avait* (Lambrecht, 1988) in (9).

(9)   [on est rentré dans un bar # qui fai-
      sait boîte]$_{du}$ $$ [y avait # que nous]$_{du}$
      # $$ [y avait la musique # à fond les
      ballons]$_{du}$
      –
      [we entered a bar # that was also a
      nightclub]$_{du}$ $$ [there was # only us]$_{du}$
      # $$ [there was music # (that was) ex-
      tremely loud]$_{du}$

## 6   Conclusion and Discussion

In this work we have compared different approaches for building a discourse unit segmenter adapted to French conversations. We had access to a manually segmented corpus of significant size which allowed us to perform a wide range of experiments. First we compared the option of (i) using our conversational data set to fine-tune an existing discourse segmenter developed and trained for written data, (ii) a data-programming approach that makes use of the same "text-based" discourse segmenter but enriched with manual defined rules (Labeling Functions). We found that both approaches yielded similar results. This suggests that both approaches are worth considering depending on the exact use case. While data-programming requires some heuristic rule engineering, fine-tuning requires annotated data that is costly to obtain, especially for relatively expert tasks such as discourse segmentation.

We also ran a second experiment to investigate (i) how much manually annotated data is required before reaching the same performance as the data-programming approach; (ii) whether starting from a written base segmentation model was useful at all (compared to

training the segmenter directly over the BERT pretrained language model). To the first question, it is notable that given the significant variability between folds, only a small amount of annotated data ($\sim 20\%$) is sufficient to get close to the best results we obtained. Moreover, starting from a written discourse segmenter model or directly from BERT did not significantly change the results.

Overall, our findings suggest that annotating (segmenting) a large amount of conversation might not be necessary since both the data-programming approach (that makes use of an existing discourse segmenter developed on written data) and a model trained with little data (here about 2700 discourse units) yielded results comparable to a model fined-tuned on our whole training set (13500 discourse units).

Breaking down the performance of the different models, we see that both the fine-tuned model and the weakly supervised one improve over the pause baselines. While pauses are strong predictors (rather high precision for a baseline), many discourse units are not preceded by a pause, so extra cues are needed. Both models seem to easily learn how to segment within "fluent monologues" (even without pauses)—an result likely explained at least in part by the role of the existing discourse segmenter and the relevant language model. However, when speech becomes strongly disfluent,[4] in particular when disfluency gets tangled up with discourse markers that typically signal discourse segment starts, both approaches struggle. Finally, certain constructions, such as presentatives, which are frequent in conversational language but absent from written data, are also an issue. Overall while the models are definitely useful as discourse segmenters, their scores are way below the state-of-the-art obtained on written text. Apart from the fact that the task of EDU segmentation is arguably more difficult for spoken language, underlying biases carried by segmenters trained on written data could explain in part why our models remain relatively confused when facing token sequences found only in conversational data sets, despite fine-tuning

or our attempt to add heuristic specific rules.

As future work, we plan to refine our experiments by separating discourse units into two categories: easy and difficult to segment. Indeed, in conversation, a sizeable amount (about 19%) of discourse units are trivial to segment (single or lexical feedback items preceded and followed by long pauses) while some others, as we have seen in error analysis, are really complex to delineate. Our opinion is that separating these two cases at all stages (including inter-coder agreement measures) will allow us to learn more about discourse segmentation of conversation and ultimately help in developing better performing models.

## Ethics Statement

This paper does not process any sensitive material and does not generate any content. It does not raise any ethical issues.

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

## A  Useful token list for Labelling Functions

- BEGIN_TOK = DM_INI + PRO_SUJ + FEEDBACK

- ENDING_TOK = quoi,hein

- NON_ENDING_TOK = PRO_SUB + PRO_OTH + PRO_REL + DEM + DET + PREP + OTHER + NEG + DM_INI

with the following lists:

- PRO_SUB = je, tu, il, vous, on, nous, elle, ils, elles, j',c',t',y

- PRO_OTH = me, te, se, mes, tes, ses, mon, ton, son, ma, ta, sa, nos, vos , leur ,ceux

- PRO_REL = qu', que, qui, quel

- DEM = ce, cette, ces, cet

- DET = le, la, les, un, une, des , l', d'

- PREP = à, de, par, pour, en, dans, chez, sur, sous, pendant, avec

- OTHER = soit, juste, pendant, surtout, chaque, quelque, quelques, sauf

- NEG = n', ne

- DM_INI = mais, donc, parce, ah, alors, c'est-à-dire, puisque, bah

- FEEDBACK = mh, ouais, ah, oui, bon, voilà, putain,oh, okay,ok,euh,ben,et,d'accord, non

## B  Part-of-Speech list for Labelling Functions

- NON_ENDING_POS = DET, CCONJ, SCONJ, ADP

- ENDING_POS = INTJ

- BEGINNING_POS = INTJ, CCONJ, SCONJ

- NON_BEGINNING_POS = VERB, AUX

- NO_RULES = ADJ, NOUN, ADV, NUM, PROPN, X