# OpenReview forum: "Comparing Methods for Segmenting Elementary Discourse Units in a French Conversational Corpus"
_NoDaLiDa/2023/Conference — NoDaLiDa 2023_

### Official Review · Reviewer_NXVy · 2023-03-09
**Good paper contrasting approaches to training a discourse segmenter for French dialogue, at some places hard to follow**

**Rating:** 7
**Confidence:** 3

**Review:**

The authors compare different methods for training/inducing a discourse segmenter for French dialogue: fully supervised fine tuning an existing system, training on restricted amounts of data, and using data programming. It is overall a well executed paper that although it doesn't introduce new techniques does further our understanding of how these techniques may be applied and what there effectiveness is.

- The inclusion of `tony-w` in the data programming approach makes __a lot__ of sense from the perspective of getting the most out of the available resources and demonstrating the flexibility of the data programming approach, **however** it muddles the picture when looking at the effectiveness of the data programming strategy on its own: a researcher looking to use these results as a starting point to form an idea of which approach to use in a new tool construction effort for a task that is slightly different or a language that doesn't already have tools available will get an overoptimistic view of data programming, since the results here "hide" a complete previously developed system. Perhaps one can have ones cake and eat it, too, by reporting two scores for the data programming approach: one with tony and one without. (It is to be expected that the one without tony will fare worse and the balance will tip further in favour of annotating a little data and against data programming in that case.)

- It would be good to get some info about the distribution of labels in the data: what %age of token boundaries is also DU boundaries? I see the total number of DUs in the CID, but not the number of tokens, the number of conversations, etc.

- I was a bit confused by the remarks around l275, because I thought this meant all of the tested systems could only predict a discourse boundary  when there was a break of at least 200ms, but from reading the rest of the paper, that is wrong (there are for instance systems that have higher recall than they .2 baseline).

- The comparison of systems in terms of f-scores is misleading. Fortunately the authors do talk in enough detail about the systems and their performance to counteract this somewhat, but still, when the authors in the conclusion write

  > we compared the option of (i) fine-tuning on our conversational data set an existing discourse segmenter developed and trained
  > for written data, (ii) a data-programming approach that makes use of the same “text-based” discourse segmenter but enriched
  > with manual rules (Labeling Functions). We found that both approaches yielded similar results.

  they fall in the f-score trap. Yes, the f-scores __are__ similar, but the obtained systems (which must the understood as the meaning of "results" in the quote above?!) behave very differently, as can be seen by looking at the precision and recall scores in figures 2 and 3. tony_w_ft does slightly better in terms of precision than in terms of recall, but the scores are close, data programming does a lot better in terms of recall than in precision with a big gap. (In fact the improvement finetuning brings to tony_w appears to be largely interms of a great gain in precision at the cost of only a slight drop in recall.) The curves in figure 5 also only show f-score and therefore lets one think one gets very similar systems at the 20% mark, but this need not be the case at all. The curves for the finetuned tony and the nomodel BERT also look similar when just looking at f-score, but here, too, very different models may be hiding.

- Around  l 308: "finally a supervised model is trained on the dataset that had been annotated by the label model". Which (kind of) supervised model is this? Is this something you chose or is it part of the specification of the Snorkel paradigm? I can imagine that this choice may have a big influence on the outcome.

- Despite my remarks about the focus on f-score above, I found the discussion of the results and the error analysis overall well-written and insightful!

- A small layout request: the distance between the figures and the discussion in the text is pretty big (several pages at the worst), please see if the layout can be rearranged to improve this situation.


**Paper Type:**

Long paper

---

### Official Review · Reviewer_yq2m · 2023-03-14
**A comparison of different methods for discourse segmentation in French**

**Rating:** 6
**Confidence:** 3

**Review:**

The paper describes an evaluation of different types of models for discourse segmentation of spoken dialogues in French.

Two types of systems are compared: a hand-engineered set of rules, and the fine-tuning of an existing ML-based discourse segmented on the specific dataset used here. Different training approaches for the ML-based system are compared: starting from a BERT model or starting from the previously trained discourse segmenter. The main takeaway seems to be that the differences between systems are rather small for this task. With a small amount of data, the rule-based approach works better, while the ML-based solution catches up when enough data is available. Fine-tuning an existing segmenter seems to be roughly comparable to fine-tuning a BERT model.


**Paper Type:**

Long paper

---

### Official Review · Reviewer_Uhez · 2023-03-15
**Comparing Methods for Segmenting Elementary Discourse Units in a French Conversational Corpus**

**Rating:** 7
**Confidence:** 3

**Review:**

Comparing Methods for Segmenting Elementary Discourse
Units in a French Conversational Corpus

In this paper, the authors present a comparative analysis of various methods for segmenting conversational French speech.  In particular, they i) fine-tune a discourse segmenter pre-trained for written data, and (ii) enrich a sota data-programming approach with manual rules.

The topic, goal, and the motivation of the work, are clear and well-defined. The paper is well written and the technical quality is appropriate. Different experiments and setup are cleary motivated and discussed.

However, a careful review of syntax is required to drop typos. For example:
- row 79, term -> termS
- row 113, became -> become
- row 401, exhibits -> exhibit
- row 483, was
- row 528, or -> of
- row 743, problem -> problems
...

As a further revision, the authors should maintain consistency in their use of English (US or UK). For example, along the paper both labelled/ing and labeled/ing are used.

**Paper Type:**

Long paper

---

### Decision · Program_Chairs · 2023-03-17

Accept